# Research on MEMS Solid-State Fuse Logic Control Chip Based on Electrical Explosion Effect

**DOI:** 10.3390/mi14030695

**Published:** 2023-03-21

**Authors:** Wenting Su, Wenzhong Lou, Hengzhen Feng, Yuecen Zhao, Sining Lv, Wenxing Kan, Bo He

**Affiliations:** 1Science and Technology on Electromechanical Dynamic Control Laboratory, School of Mechatronical Engineering, Beijing Institute of Technology, Beijing 100081, China; 2National Key Laboratory of Science and Technology on Micro/Nano Fabrication, Shanghai Jiao Tong University, Shanghai 200240, China; 3School of Electronic Information and Electrical Engineering, Shanghai Jiao Tong University, Dong Chuan Road 800, Shanghai 200240, China

**Keywords:** MEMS, solid-state fuse logic control chip, metal bridge, solid-state ON–OFF switch

## Abstract

A microelectromechanical systems (MEMS) solid-state logic control chip with three layers—diversion layer, control layer, and substrate layer—is designed to satisfy fuse miniaturization and integration requirements. A mathematical model is established according to the heat conduction equation, and the limit conditions of different structures are presented. The finite element multi-physical field simulation method is used to simulate the size and the action voltage of the diversion layer of the control chip. Based on the surface silicon process, fuse processing, and testing with the MEMS solid-state fuse-logic control chip, a diversion layer constant current, maximum current resistance test, and a control layer of different bridge area sizes, the bridge area size is 200 × 30 μm, and the minimum electrical explosion voltage is 23.6 V. The theoretical calculation results at 20 V and 100 μF demonstrate that the capacitor energy is insufficient to support the complete vaporization of the bridge area, but can be partially vaporized, consistent with the experimental results.

## 1. Introduction

Reducing the volume of fuse security systems is a primary research direction for fuses. Among the major western military powers, there are two predominant ideas for fuse miniaturization: (1) microelectromechanical systems (MEMS) research on fuse core components conducted by major research institutions, such as the US Army and Navy, and (2) solid-state and hybrid integration research of fuse core functions conducted by some research institutions in Europe, France, and Germany. In response to these ideas, there are currently relevant principle prototypes, among which, the US military has higher technical maturity and richer types. It has been used in the Army, Navy, and other ammunition.

In 2001, Charles H.Robinson [1,2] of the United States Armament Research, Development, and Engineering Center (ARDEC) proposed a MEMS security device driven by inertial force and applied it to the American ideal individual weapon (OICW) 20 mm high explosive grenade set distance air blast fuze [3]. In 2010, a layered PyroMEMS security device was proposed by the Laboratory for Analysis and Architecture of Systems (LAAS) of the French Centre national de la recherche scientifique (CNRS) [4,5]. Xingdong Lv [6] proposed a large-displacement MEMS electromagnetic driver in 2015. In 2016, Taylor T. Young [7] proposed a MEMS security device driven by mechanical environmental force and electric heat. In 2016, Xiuyuan Li [8] proposed a linear high-speed electric micro-actuator for MEMS security devices. The device was composed of a micro-spring, a diaphragm, and four V-shaped electric micro-actuators with a micro-lever amplification mechanism. In 2017, Hu Tengjiang [9] from Xi ‘an Jiaotong University proposed an electric heating drive security device composed of four electric micro-actuators and micro-levers.

The fuse miniaturization study of the French army established a new direction, redefined the process of fuse safety, guaranteed release and explosion, and integrated the solid-state control device into a chip. The study integrated the component with a mechanical insurance slider based on precision machining, forming a fuse core component with high security. In contrast to MEMS technology, the French technology has higher maturity; devices from the related research institutes are more advanced and smaller, but challenging to implement.

In 2007, the French LAAS first conducted two different structures of the MEMS ON–OFF solid-state fire switch study [5,10,11], which can be applied to the MEMS fuse. In 2011, Baginski [12,13] in the US proposed a high-pressure gas gap actuator that provides a low-cost alternative to the traditional triggering spark and is applied to the initiation system containing exploding foil initiator (EFI) detonators. The actuator substrate is ceramic, deposited with metal, and graphically transformed into a solid-state actuator. However, the actuator has not been widely used because the driving voltage is too high. In 2012, the Beijing Institute of Technology designed a solid-state actuator based on an electrical explosion [14]. The structural design of the actuator was conducted through simulation and processing. The processed electrical explosion device was a solid-state actuator, and the metal was aluminum (Al); the high resistivity of Al can ensure that the processed actuator can work reliably. A high-energy capacitor actuator can operate reliably under working conditions of 40 V and 220 μF.

In 2016, the Beijing Institute of Technology proposed a solid material using epoxy resin as a switch to create a solid switch [15]. In 2017, India implemented a circular switch [16] with trigger electrodes on an alumina substrate consisting of two printed arc electrodes and a thin strip used as a trigger electrode, which was easily integrated with electronic devices for ignition applications. In 2019, the Beijing Institute of Technology proposed a MEMS miniature electrostatic dredge based on the Paschen’s law size effect [17,18] and adopted the design idea of an electrostatic dredge charge. They used MEMS technology to design a micro-discharge electrode group with a reliable breakdown at a low voltage threshold, and a release gathered between the pins or pins and shell and dredge flowed through an electrostatic component. Finally, they conducted radio frequency and other adverse environmental stimulations and improved firearms safety.

Based on the electrical explosion theory, this study designs a fuse ON–OFF solid-state switch control chip based on the COMSOL finite element simulation and verifies that the chip has a high flow resistance, lower trigger voltage, and guaranteed fuse safety logic control.

## 2. Modeling

### 2.1. Fuse-Logic Control Chip Function and Unit Structure

Based on the ON–OFF solid-state switch of the indirect electrical explosion effect, this study designs a fuse logic control chip with four states, as depicted in Figure 1. Switches 1, 2, 3, and 4 are solid-state ON–OFF switches.

When the fuse is in a safe state, the main components in the fuse core-initiation control system are short-circuited (Figure 1a). When the power supply is powered on, the short circuit insurance at both ends of the detonating capacitor (1) is automatically released, and the capacitor is charged (Figure 1b). According to the preset conditions, under the control of the external power control signal, the short circuit insurance at both ends of the detonator (4) is released, and the fuse enters the ready state (Figure 1c). When the initiation control signal arrives, if the initiation control system operates normally and the fuse is in a good condition, then the initiation is normal (Figure 1d). If the fuse is not normally detonated, the switch on the main channel of the ignition energy is disconnected to ensure the safety of the unexploded dumb bomb (3), the main channel of ignition energy is broken, and the detonator cannot obtain the safety of detonation energy recovery (Figure 1e). The switch (2) connecting the energy storage capacitor is disconnected, so that the energy storage capacitor is disconnected from the power supply to ensure the safety of unexploded dumb bombs (Figure 1f).

Furthermore, if the electrical insurance or energy control or reliability fails in any given state, the fuse shall be designed to prevent the premature release of the insurance (start) or premature action, including the incidental and dependent failures occurring before, during, or after the application of the fuse power supply.

The solid-state ON–OFF switch of the fuse-logic control chip is a sandwich structure containing two main functional layers: the flow guide layer and the control layer. The diversion layer is located above the control layer, and its resistance value is far less than that of the pyrotechnics, which can channel the current flowing through the pyrotechnics. When applied, it can directly connect with the pyrotechnics in parallel, improving the safety current and ignition energy threshold of the pyrotechnics. The control layer is used to release the diversion layer, and the external control circuit is connected when applied. Under the action of the external control circuit, the electrical explosion of the control layer will disconnect the diversion layer above it and restore the ignition energy threshold of the explosive product. In this paper, the mode of electrical explosion in the control layer to disconnect the diversion layer above the control layer is called the indirect electrical explosion effect. Figure 2 illustrates the top view and profile of the basic functional unit of the solid-state ON–OFF switch of the fuse-logic control chip.

When the solid-state ON–OFF switch based on the indirect electrical explosion effect operates, the diversion layer is in the normally open state, and the flow resistance of the diversion layer is greater than the ignition threshold value of the initial ignition product. After the action of the control layer, the diversion layer is disconnected, and the ignition threshold of the initial ignition product is restored. Throughout the process, it is necessary to ensure that the diversion layer can withstand a large current when not disconnected; the explosion control layer can quickly explode at low voltage to disconnect the diversion layer. The circuit is depicted in Figure 3.

### 2.2. Switch Simulation Based on the Electrical Explosion Effect Solid-State ON–OFF

The control-layer energy-exchange element of the solid ON–OFF switch based on an electrical explosion is a metal bridge. Because of the high temperature and shock wave during the electrical explosion, the metal wire layer on the surface of the bridge area is destroyed under the action of high temperature and shock wave, and the metal wire layer is disconnected. Thus, the metal of the switching wire layer is Al.

The solid model and simulation results are depicted in Figure 4. The Al wire surface is covered by the silica passivation layer, and the convective heat transfer conditions are set on the surface of the silica passivation layer, metal bridge layer, and SiO_2_. The layer thickness of the insulation layer and the Al wire is 1 μm, and the open boundary condition is set between the surface of the silicon substrate and the surface air boundary. The Joule heat module in COMSOL is used for transient simulation analysis. The excitation source adopts the form of a capacitance discharge. The selection of the discharge voltage here must also ensure that the current density in the metal bridge is greater than the previously calculated value, or more than approximately 10^7^ A/cm^2^.

### 2.3. Simulation of Flow Resistance of Solid-State ON–OFF Switch Diversion Layer Based on Electrical Explosion Effect

The mechanism of action in the solid-state actuator of electric explosion is Joule heat, which is generated after the control layer is received by the external control signal—the temperature of the control layer melts from solid to liquid, then evaporates to gas and continues to heat up, resulting in the change of arc breakdown, and thus, the electric explosion occurs. In the process of electric explosion, metal heat accumulation and temperature rise are mainly affected. The factors affecting the occurrence time of electric explosion include the current and resistivity of metal materials. In the process of electric explosion, the meaning represented by the amount of analogy is the Joule heat formed on the unit resistance of the current energy in the unit area of the metal bridge area on the cross section cut off in the direction of the current flow at a certain time. The comparison amount of metal bridge area can be defined by the following formula:(1)$$g_{ex}=\int_{0}^{t}j_{ex}^{2}dt$$
where $g_{ex}$ represents the specific action value of the metal bridge area, $j_{ex}$ represents the current density flowing through the metal bridge area, and *t* represents the ongoing time. According to the electric explosion effect, the complete process of the metal bridge area can be divided into six stages, as shown in Figure 5.

Where the resistivity of the metal bridge can be expressed as:(2)$$\rho_{ex}=\frac{\rho_{ex1}}{\sqrt{1-\frac{\rho_{ex2}^{2}-\rho_{ex1}^{2}}{\rho_{ex2}^{2}}}\cdot \frac{g_{ex\max}}{g_{ex}}},\;0\leq g_{ex}\leq g_{ex\max}$$
where $\rho_{ex}$ represents the resistivity of the metal bridge area, and $\rho_{ex1}$ and $\rho_{ex2}$, respectively, represent the resistivity of the solid and liquid phases of the metal bridge.

Phase I: Solid heat absorption reaches melting point in metal bridge zone of control layer.
(3)$$Q_{I}=\int_{t_{0}}^{t_{M1}}i_{t}{}^{2}R_{exs}dt$$
where $t_{M1}$ is the time when the metal bridge begins to melt, and $R_{exs}$ is the initial resistance of the metal bridge area.

Phase II: Solid melts to liquid, and the endothermic equation of bridge zone is:(4)$$Q_{\mathrm{II}}=\int_{t_{M1}}^{t_{L1}}\left((i_{M1}e^{\frac{t-t_{M1}}{R_{esl}}}{)}^{2}R_{exs}-\frac{\lambda_{ex}}{d}Nu\left(T_{L1}-T_{M1}\right)A_{es}t\right)dt$$
where $T_{L1}$ is the temperature at which the metal bridge is completely liquefied; $T_{M1}$ is the temperature at which the metal bridge begins to melt; $R_{esl}$ is the resistance of the metal bridge during liquefaction; $i_{M1}$ is the current during liquefaction; $t_{L1}$ is the time of the metal bridge during liquefaction; $A_{es}$ is the area of the metal bridge.

Phase III: Liquid temperature rises in the metal bridge zone of the control layer, and the heat absorption equation of the bridge zone is:(5)$$Q_{\mathrm{III}}=\int_{t_{L1}}^{t_{V1}}\left({\left(i_{t}e^{\frac{t-t_{L1}}{R_{esl}}}\right)}^{2}R_{exL}-\frac{\lambda_{ex}}{d}Nu\left(T_{V1}-T_{L1}\right)A_{es}t\right)dt$$
where $t_{V1}$ is the temperature of heating to gasification.

Phase IV: Bridge area gasification and electrical explosion
(6)$$Q_{\mathrm{IV}}=\int_{t_{V1}}^{t_{G1}}\left({\left(i_{t}e^{\frac{t-t_{G1}}{R_{exg}}}\right)}^{2}R_{exg}-\frac{\lambda_{ex}}{d}Nu\left(T_{G1}-T_{V1}\right)A_{es}t\right)dt$$
where $T_{G1}$ is the temperature when the metal bridge is fully vaporized, $t_{G1}$ is the time when the metal bridge is fully vaporized, and $R_{exg}$ is the resistance when the metal bridge area is vaporized.

The transient simulation analysis was performed using the Joule thermal module in COMSOL. The model verifying the maximum flow tolerance of the switching diversion layer is depicted in Figure 6.

The simulation results of the diversion layer are depicted in Figure 7. A constant current source circuit is selected for simulation. The metal bridge in the switch reaches the vaporization temperature and the switch function. The switch bridge area is adjusted, and different currents are applied to verify the maximum current value when the metal bridge area reaches the melting point. The width of the metal bridge area is set as 30, 40, 60, and 70 µm, and the current values that can be tolerated are 1600, 2000, 2600, and 3000 mA. When the width of the metal bridge is increased, the resistance of the bridge area decreases, and the flow resistance increases.

Reducing the resistance of the bridge area can increase the flow resistance of the solid-state actuator, according to the calculation of the mathematical model of heat accumulation and temperature rise. Therefore, the two-bridge model is used for simulation analysis to ensure the safety of the solid-state actuator and increase its flow resistance. Two groups of bridge areas are selected—the 60 × 70 µm and 60 × 60 µm bridge areas, with different structures. The temperature isotherm diagram of bridge areas is depicted in Figure 8.

Increasing the flow resistance of the bridge area increases the action voltage, safety, and reliability of the pyrotechnics. In Figure 1c, switch 3 can pass through a higher current, that is, it can increase the ignition voltage of the pyrotechnics. As depicted in Figure 7, when the current value is 5400 mA, the temperature rises to the temperature at which the material can undergo phase transformation and remain stable. In comparison, when the current value is 5000 mA, the material phase transformation temperature can be reached and remains stable. Therefore, the two-bridge zone model can significantly increase the flow resistance. Furthermore, the action law of the flow resistance in the double bridge area is the same as that in the single bridge area. Increasing the width of the bridge area can increase the flow resistance.

### 2.4. Control Layer Simulation

Another critical performance measure of a solid-state ON–OFF switch based on the electrical explosion effect is response time—the control layer action time is the electrical explosion occurrence time. For determining the response time of the solid-state actuator designed in this study, a multi-physical field simulation is conducted on the process of the solid-state actuator control layer heating up to an electrical explosion under the action of the current, and the control layer is driven by capacitor discharge. The peak voltage is 10 V, and the energy storage capacitance is 33 μF. The temperature rise of the 60 × 60 µm bridge area is simulated. The temperature curve of the bridge area with time when the electrical explosion occurs is depicted in Figure 9. Four points in the bridge area were selected symmetrically as the temperature research objects.

Figure 9 is the simulation result. The figure on the right shows the temperature curve of four points in the bridge area with time. When a certain current is applied to the bridge area, its temperature will increase rapidly with time. Therefore, the time when the temperature reaches the maximum value is the time when the electric explosion occurs. Based on analyzing the warming process in the 60 × 60 µm bridge area, the results of the specific temperature of the region simulated with time are depicted in Figure 10.

As depicted in Figure 10, the bridge area’s temperature increases rapidly with time. At 1.75 μs, the control layer reaches the melting point and gradually begins to melt (aluminum has a melting point of 660 °C and a boiling point of 2367 °C). In the simulation model, the temperature of the control layer continues to rise when the current continues to be applied, and the control layer begins to vaporize after another 6.7 μs. When 57.15 μs is reached, an electric explosion occurs, and effective action is realized. As demonstrated by the simulation results, if the solid-state ON–OFF switch based on the electrical explosion effect is to function reliably, it must ensure that the temperature can rise rapidly at a given time. Furthermore, during the design process, it must ensure that the electrical explosion occurs in the middle of the bridge area.

Based on this analysis, the solid-state ON–OFF switch model parameters designed in this study are presented in Table 1.

## 3. Fabrication

Based on the simulation results, the fuse logic control chip designed using Ledit is depicted in Figure 11.

The fuse-logic control-chip design includes four solid-state ON–OFF switches with different thresholds, which are used to test the flow resistance of different sizes of the diversion layer bridge area and the electrical explosion time of the control layer. The specific structural parameters of the bridge area in the switch design map from Figure 11 are presented in Table 1. The processing process of solid-state switch is shown in Figure 12.

Figure 13 illustrates the fuse-logic control chip after processing.

## 4. Test and Discussion

### 4.1. Constant Current Dredging Ability Test

As depicted in Figure 11, the circle-level test system includes the probe station, multimeter, virtual oscilloscope, pulse voltage generator, and constant current source. The probe station is used to load the drive signal and constant current on the circle, the multimeter is used to measure the resistance of the diversion and control layers, the virtual oscilloscope is used to test the voltage and current signal flowing through the diversion and control layers, the constant current source is used to provide the constant current to the diversion layer, and the pulse voltage generator is used to provide the pulse voltage to the control layer.

The constant current dredging ability test measures the current resistance of the diversion layer. During the test, the constant-current source is connected to the diversion layer to adjust the constant current source output current. If an electrical explosion occurs in the diversion layer, the current value is the maximum acting current value of the switch designed in this study. The test system is shown in Figure 14. The test results are depicted in Figure 15.

As depicted in Table 2 and Figure 16, the measured resistance value is typically higher than the theoretical resistance value because the theoretically calculated resistance value does not include the welding pad resistance, trapezoidal resistance, or bonded wire resistance. Figure 16a represents the error distribution histogram of the resistance of each group of four structures, Figure 16b represents the error distribution histogram of the maximum current of the control layer of each group of four structures, and the shaded part represents the size of the error value.

Furthermore, the possible thickness loss caused by the processing process error will also lead to an increase in resistance. These phenomena also produce an error between the measured and theoretical values of the dredging current; the overall error is less than 10%, indicating the correctness of the theoretical calculation and simulation.

### 4.2. Control Layer Initiation Energy Test

The electrical explosion test in the control layer is continued, as presented in Table 3. Based on the results, if the control layer of a specific size is electric exploded, the lower the energy storage capacitance, the higher the applied voltage. The control layer designed in this study can be operated under the energy supply condition of the fuse to ensure that the short circuit state of the explosive product, and the capacitor can be removed according to the established function.

## 5. Conclusions and Perspectives

In this study, a solid-state fuse-logic control chip based on an electric-explosion-effect solid-state ON–OFF switch is designed. Based on a simulation of the solid-state ON–OFF switch using the COMSOL multi-physics field, the maximum current-resistance value and minimum breakdown voltage of the solid-state ON–OFF switch of the Al bridge are determined to verify the operating condition shell action reliability of fuse-logic control chip. The size of the ON–OFF solid-state switch diversion layer of the Al bridge is set as 30 × 60, 60 × 60, and 60 × 100 µm. The current values that can be tolerated by 30 × 60, 60 × 60, and 60 × 100 µm are 1600, 2600, and 4000 mA. The solid-state logic controller was processed; solid-state ON–OFF switches of the 30 × 60, 60 × 60, 60 × 100, and 70 × 60 × 2 µm sizes of the diversion layer were tested. Without considering the resistance of the pad, the difference between the test and theoretically calculated values and the simulation value was less than 10%. The driving energies of the 60 × 60, 100 × 30, and 200 × 30 µm control layers were tested, and the results were consistent with the simulation value, confirming the theoretical calculation. A solid-state fuse-logic control chip based on an Al bridge solid-state ON–OFF switch can achieve fuse initiation, self-failure, and self-failure functions.

## Figures and Tables

**Figure 1 micromachines-14-00695-f001:**
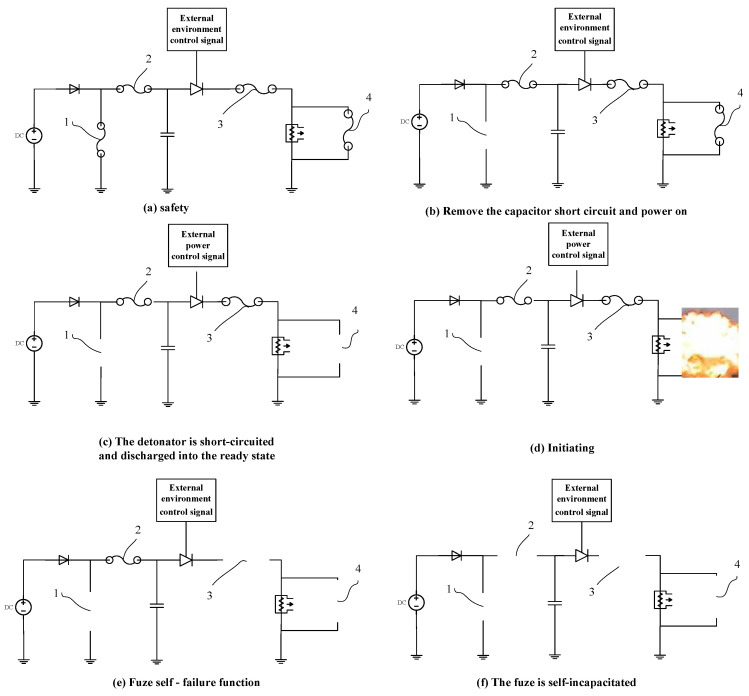
Fuse-logic control-chip deprotection logic and self-failure function.

**Figure 2 micromachines-14-00695-f002:**
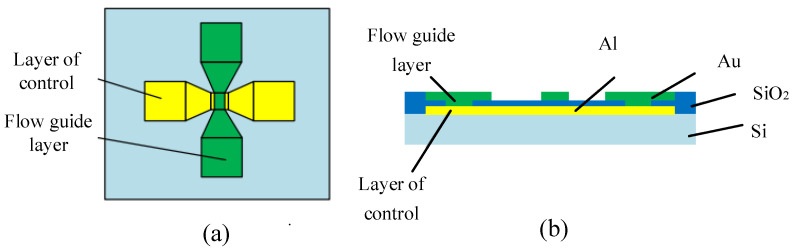
Fuse-logic control chip solid-state ON–OFF switch basic functional unit. (**a**) Top view of the solid-state ON–OFF (**b**) Bridge section view of the solid-state ON–OFF.

**Figure 3 micromachines-14-00695-f003:**
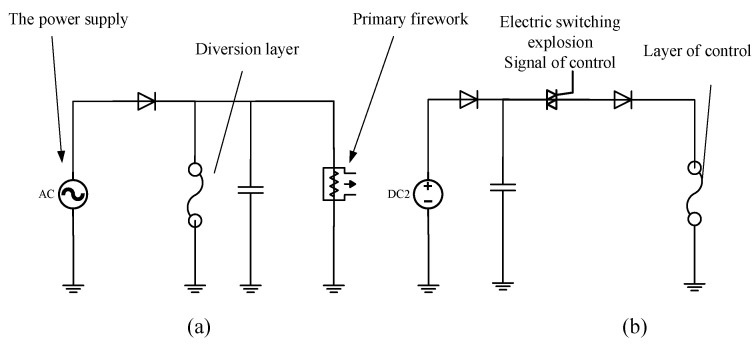
Circuit diagram of indirect electrical explosion action. (**a**) The control circuit of the explosive initiation, the diversion layer is in the normally open state. (**b**) Bridge section view of the solid-state ON–OFF.

**Figure 4 micromachines-14-00695-f004:**
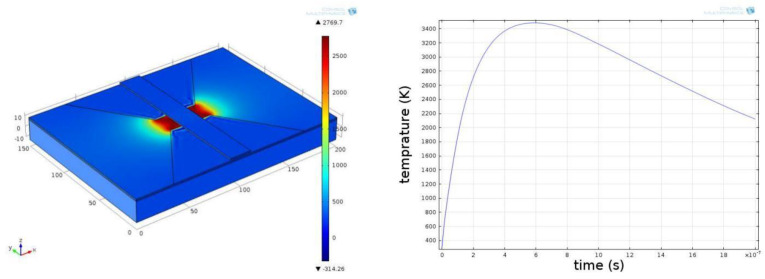
Simulation results of solid-state ON-OFF switch based on electrical explosion.

**Figure 5 micromachines-14-00695-f005:**
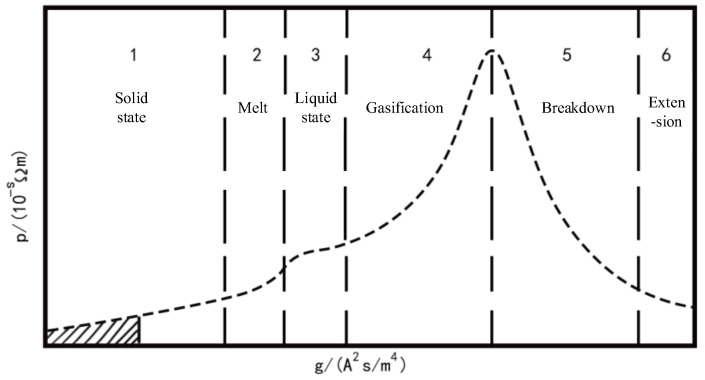
Control layer metal bridge electrical explosion process.

**Figure 6 micromachines-14-00695-f006:**
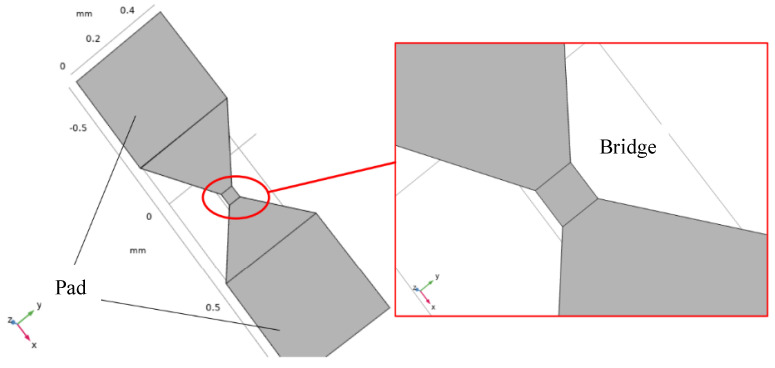
Solid-state ON-OFF multi-physical field simulation model.

**Figure 7 micromachines-14-00695-f007:**
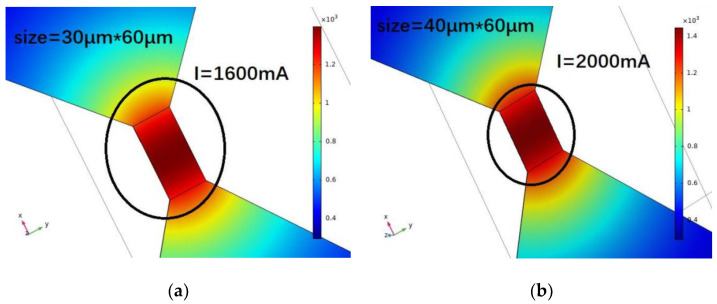
Multi-physical field simulation results of solid-state ON–OFF switch diversion layer in single bridge area. I = 1600 (**a**); 2000 (**b**); 2600 (**c**); 3000 (**d**) mA.

**Figure 8 micromachines-14-00695-f008:**
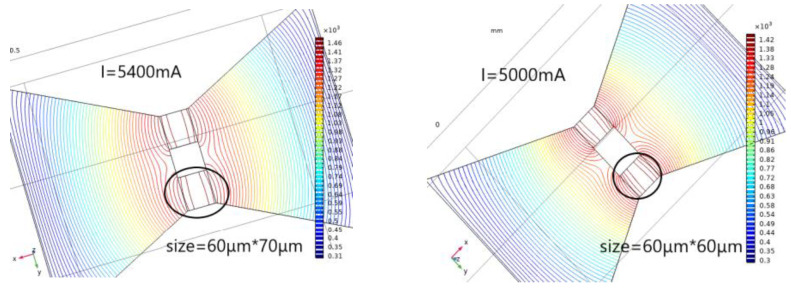
Multi-physical field simulation results of solid-state ON–OFF switch diversion layer in double bridge area.

**Figure 9 micromachines-14-00695-f009:**
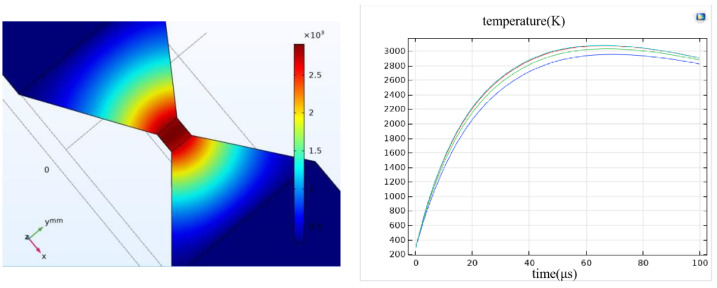
Solid-state ON–OFF switch controls the temperature rise curve of bridge area with time.

**Figure 10 micromachines-14-00695-f010:**
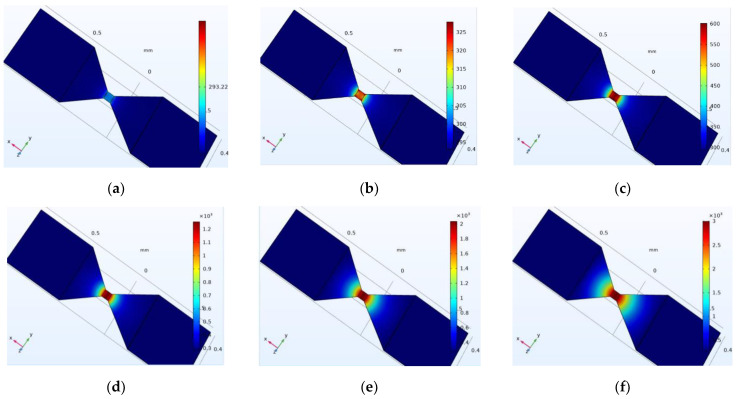
Temperature rise in bridge area of solid-state ON–OFF switch control layer changes with time. (**a**) time = 0 μs, (**b**) time = 0.15 μs, (**c**) time = 1.75 μs, (**d**) time = 6.7 μs, (**e**) time = 27.05 μs, (**f**) time = 57.15 μs.

**Figure 11 micromachines-14-00695-f011:**
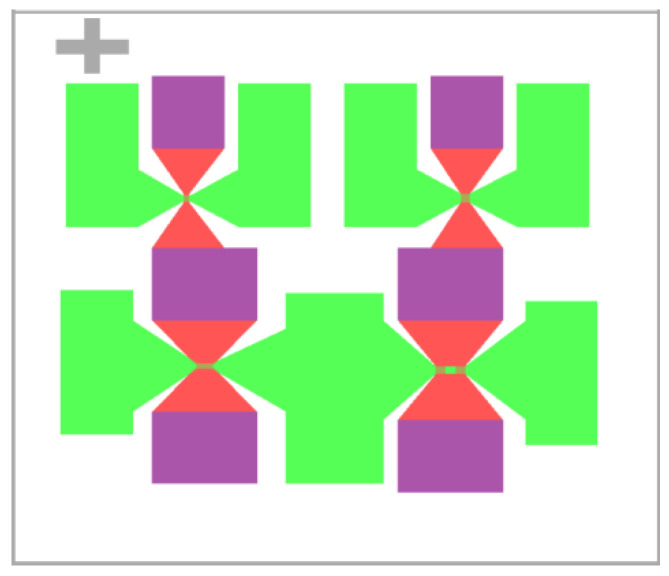
Fuse-logic control chip.

**Figure 12 micromachines-14-00695-f012:**
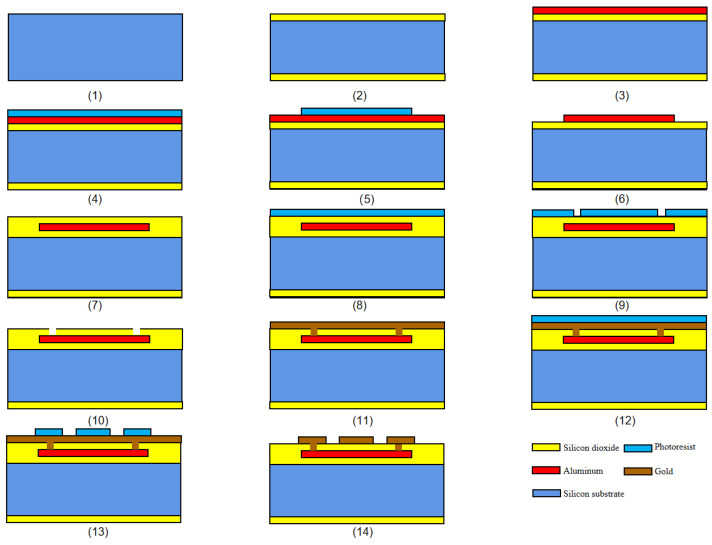
Fuse-logic control-chip process flow: (**1**) The substrate selects 400 μm <100> crystal double-throwing silicon wafer. (**2**) First, the thermal oxidation process is conducted to obtain a 400 nm thick silicon oxide layer on the front and back of the substrate to ensure insulation between the device structures. (**3**) Suttering 400-nm thick Au as the control layer. (**4**,**5**) Photoresist is applied, and exposure and development are removed to form a graphical area of the control layer structure. (**6**) Etch the Al control layer to create the graphical driving structure layer. (**7**) SiO_2_ insulation layer was obtained by PECVD process. (**8**,**9**) Photoresist is applied, and the exposure and development are removed to form the through-hole graphical area. (**10**) Etching SiO_2_ insulation layer; window through the hole. (**11**) Suttering 600-nm thick Au as the diversion layer. (**12**,**13**) Photoresist is applied, and exposure and development are removed to form a graphic area of the diversion layer. (**14**) Etch the guide layer Au to create the graphical diversion layer.

**Figure 13 micromachines-14-00695-f013:**
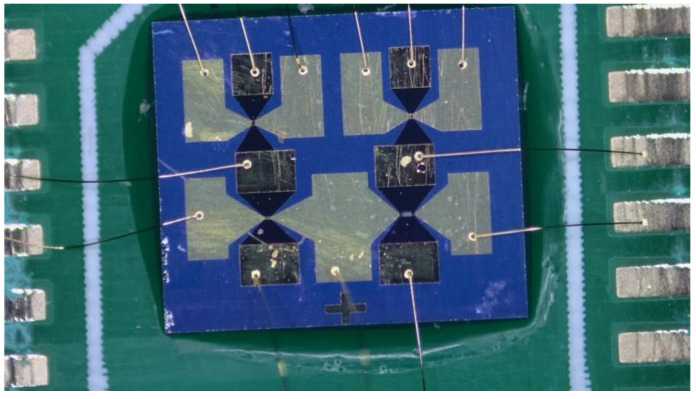
Fuse-logic control chip.

**Figure 14 micromachines-14-00695-f014:**
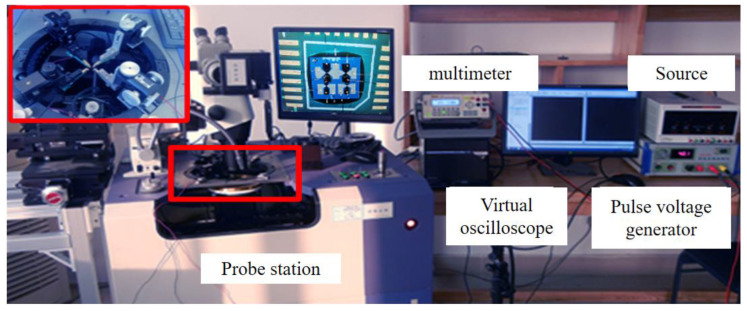
Circle-level test system.

**Figure 15 micromachines-14-00695-f015:**
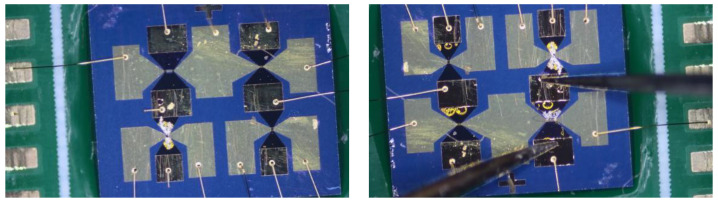
Circular level test results of different sizes.

**Figure 16 micromachines-14-00695-f016:**
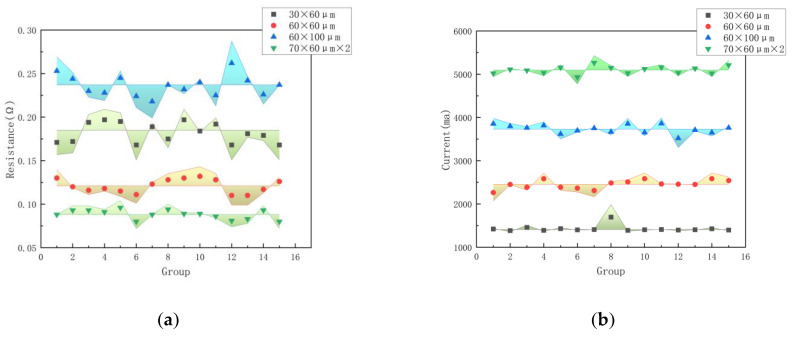
(**a**) Resistance error analysis. (**b**) Current error analysis.

**Table 1 micromachines-14-00695-t001:** Structure sizes of solid-state ON–OFF switch.

Group	Diversion Layer	Size (µm)	Control Layer	Size (µm)
1	rectangular	30 × 60	rectangular	60 × 60
2	rectangular	60 × 60	rectangular	60 × 60
3	rectangular	60 × 100	rectangular	200 × 30
4	double rectangle	70 × 60 × 2	rectangular	100 × 30

**Table 2 micromachines-14-00695-t002:** Resistance and maximum constant current value of different bridge area sizes.

Order Number	Structure 1 with the Maximum Constant Current/mA		Structure 2 with the Maximum Constant Current/mA		Structure 3 with the Maximum Constant Current/mA		Structure 4 with the Maximum Constant Current/mA	
	Resistance Value/Ω	Maximum Constant Flow/mA	Resistance Value/Ω	Maximum Constant Flow/mA	Resistance Value/Ω	Maximum Constant Flow/mA	Resistance Value/Ω	Maximum Constant Flow/mA
1	0.171	1423	0.13	2265	0.253	3856	0.088	5021
2	0.172	1386	0.12	2453	0.244	3798	0.093	5113
3	0.194	1460	0.116	2388	0.23	3762	0.093	5089
4	0.197	1392	0.118	2585	0.228	3816	0.091	5032
5	0.195	1431	0.115	2388	0.245	3620	0.096	5156
6	0.168	1403	0.111	2367	0.224	3698	0.08	4935
7	0.189	1411	0.123	2312	0.218	3752	0.088	5265
8	0.175	1698	0.128	2487	0.237	3668	0.094	5149
9	0.197	1392	0.13	2510	0.232	3856	0.089	5023
10	0.184	1406	0.132	2587	0.24	3654	0.089	5122
11	0.192	1412	0.128	2465	0.225	3862	0.086	5159
12	0.168	1399	0.11	2459	0.262	3520	0.081	5032
13	0.181	1406	0.11	2454	0.242	3712	0.083	5135
14	0.179	1429	0.117	2586	0.226	3652	0.093	5021
15	0.168	1399	0.126	2542	0.237	3763	0.08	5210
average	0.182	1410	0.121	2457	0.236	3733	0.088	5097
theoretical	0.166	1600	0.083	2600	0.198	4000	0.065	5400

**Table 3 micromachines-14-00695-t003:** Bridge areas of different sizes.

Control Layer Structure	33 μF (V)	100 μF (V)
60 × 60 μm	12.2	5.9
100 × 30 μm	15.8	8.3
200 × 30 μm	25.3	23.6

## Data Availability

The data presented in this study are available on request from the corresponding author.

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
