# Peer review of "Research on MEMS Solid-State Fuse Logic Control Chip Based on Electrical Explosion Effect"

_micromachines, 2023, doi:10.3390/mi14030695_

Round 1

Reviewer 1 Report

The paper concerns the design of a three-layer MEMS solid-state logic control chip that must meet miniaturization and integration requirements. Theoretical and FEM models are presented to find the optimal geometry for the application requirements. Theoretical and numerical results are also compared with experimental results that confirm the predictions of the analytical-numerical models with a maximum error of 10%.

The article may be of interest to Micromachines readers but needs extensive revisions prior to publication.

Major concerns:

- The Modeling section needs to be expanded and better described. For example, the formula needed to calculate the bridge's area resistance is not given at all.
-The indirect electrical explosion effect should be better explained and a theoretical model for the calculation of the explosion voltage should be provided too.

- The symbols in Eq. 2 and 3 must all be defined within the text.

- In Section 4, Table 2 and Fig.15 you mention a comparison between experimental, theoretical and numerical results. Theoretical values are not given in Table 2, and the equations for calculating: the bridge's area resistance, the current and resistance errors are also missing.

- Line 80, p. 11 you state: "the overall error is less than 10%" but this result cannot be gained from Fig.15 or Table 2.

Other issues can be found in the attached document.

Author Response

Dear reviewers and editors,

Greetings!

First of all, thank you very much for taking time out of your busy schedule to read and revise my article. Thank you for your valuable advice. You have made a comprehensive correction to the structure, content, research methods and results of my paper. It has played a very important role in improving the quality of my thesis.

I have carefully studied the reviewer's comments and made careful modifications to the paper according to the suggestions, as follows:

  1. In view of the incomplete theoretical model you proposed, I made a complete supplement to the theoretical model;
  2. In view of the unclear experimental data you proposed, I made targeted modifications and supplements;
  3. Modified various grammar and text errors in the article;
  4. Regarding the selection of conclusion, "the readers are interested to know why you choose 10uF and 100uF. Moreover you should comment which operating condition is better (i.e., is it desiderable a lower or higher explosion Voltage? Is there a drawback/advantage in using the 100uF or 10uF capacitor? Please comment"

My reply is this "Considering the application of this design in fuze, fuze bomb damage power supply voltage is low, my reply is this" Considering the application of this design in fuze, fuze bomb damage power supply voltage is low, capacitors are usually selected 33uF and 100uF,  so the purpose of this design is to directly drive the chip on the bomb power supply, without adding booster circuit "So the purpose of this design is to directly drive the chip on the bomb power supply, without adding booster circuit"

  1. "you should indicate which is the best configuration in the Results section"

My reply is this "The optimal configuration needs to be selected according to the configuration of pyrotechnics. The. My reply is this" The optimal configuration needs to be selected according to the configuration of pyrotechnics. The control chip proposed in this paper can meet the use requirements of different pyrotechnics and is determined according  to the flow resistance value of the pyrotechnics"

6、I have put more replies in the attached price folder, please check

Finally, thank you again for your guidance and for reviewing and revising my paper again. I hope that under your guidance, I can finish this excellent paper, and I sincerely hope that my paper can be published in your journal.

Thank you and best regards.

Yours sincerely,

Wenting Su

Name: Wenzhong Lou

E-mail: louwz@bit.edu.cn

Name: Hengzhen Feng

E-mail: fenghengzhen@gmail.com

Name: Yuecen Zhao

E-mail: zhaoyc0911@sjtu.edu.cn

Reviewer 2 Report

The manuscript presents a new class of MEMS logic gates which is within the scope of the journal. In general, it is well organized, presented, and addressed the problem from a different aspect. The authors have validated their proposed design using analytical, numerical, and experimental. The fabrication process is clear. I believe the authors have to address the following comments before it gets accepted and published:

- The general scope of the paper is not clear. It must be added at the end of the introduction part. 

- Operational mechanisms need more clarity. 

- Experimental setup is missed.

- How do you plan to minimize the switching error and time?

- Sensitivity analysis is missed. 

- Figures legend and axes labels are not clear.

- Results and discussion need to be reorganized in order to show the visibility of the study.

What is the main question addressed by the research?

- This manuscript addresses a new class of MEMS logic gates.

Do you consider the topic original or relevant in the field? Does it address a specific gap in the field?

- Yes. It does. It provides a new technique for building and designing MEMS logic gates.

What does it add to the subject area compared with other published material?

- It adds fabrication techniques and simulation procedures based on an electric-explosion-effect solid state.

What specific improvements should the authors consider regarding the methodology? What further controls should be considered?

- I believe they must do more experiments to validate the visibility of the study.

Are the conclusions consistent with the evidence and arguments presented and do they address the main question posed?

- Yes. It is consistent with the manuscript.

Are the references appropriate?

- No. They can expand the literature review to cover a wide range of related studies.

Please include any additional comments on the tables and figures.

- Figures quality needs improvement.

Thank you.

Author Response

(The authors gave the same response as above.)

Round 2

Reviewer 1 Report

The revisions made by the authors significantly improved the quality of the paper. The theoretical model section is now properly presented.

I suggest aligning the labels in Figure 16 with those in Tables 1 and 2 (use "Structure 1", "Structure 2", etc.).

However, the calculation of the error between theoretical values and simulations is still not clearly explained. On Page 12, Line 480-482 you state that: "These phenomena also produce an error between the measured and theoretical values of the dredging current; the overall error is less than 10%, indicating the correctness of the theoretical calculation and simulation."

This statement should be supported by the data in Table 2. Since you do not mention which theoretical value is considered, it is not clear how the 10% value was determined. Therefore, prior to publication, the theoretical values of the resistances and currents corresponding to the 4 considered structures should at least be included in Table 2. Otherwise, the sentence quoted above is not supported by the evidence.

Author Response

Dear reviewers and editors,

Greetings!

First of all, thank you very much for taking time out of your busy schedule to read and revise my article. Thank you for your valuable advice. You have made a comprehensive correction to the structure, content, research methods and results of my paper. It has played a very important role in improving the quality of my thesis.

I have carefully studied the reviewer's comments and made careful modifications to the paper according to the suggestions, as follows:

  1. As for the "I suggest aligning the labels in Figure 16 with those in Tables 1 and 2 (use "Structure 1", "Structure 2", etc.." you suggested, the change of structure 1 in Figure 16 will cause the need to correspond to the front when reading the drawing, which will cause trouble in reading the paper;
  2. As for the error that you proposed that the current refers to 10%, the simulation in the previous table is the maximum theoretical value, here it is changed to the theoretical value;

Finally, thank you again for your guidance and for reviewing and revising my paper again. I hope that under your guidance, I can finish this excellent paper, and I sincerely hope that my paper can be published in your journal.

Thank you and best regards.

Yours sincerely,

Wenting Su

Name: Wenzhong Lou

E-mail: louwz@bit.edu.cn

Name: Hengzhen Feng

E-mail: fenghengzhen@gmail.com

Name: Yuecen Zhao

E-mail: zhaoyc0911@sjtu.edu.cn

Reviewer 2 Report

Many thanks for the updated version.

Author Response

Dear reviewers and editors,

Greetings!

Thank you again for your guidance and for reviewing and revising my paper again. I hope that under your guidance, I can finish this excellent paper, and I sincerely hope that my paper can be published in your journal.

Thank you and best regards.

Yours sincerely,

Wenting Su

Name: Wenzhong Lou

E-mail: louwz@bit.edu.cn

Name: Hengzhen Feng

E-mail: fenghengzhen@gmail.com